# The Interplay between Immune and Metabolic Pathways in Kidney Disease

**DOI:** 10.3390/cells12121584

**Published:** 2023-06-08

**Authors:** Lili Qu, Baihai Jiao

**Affiliations:** 1Division of Nephrology, Department of Medicine, School of Medicine, University of Connecticut Health Center, Farmington, CT 06030-1405, USA; 2Department of Immunology, School of Medicine, University of Connecticut Health Center, Farmington, CT 06030-1405, USA

**Keywords:** immune, metabolic, inflammation, kidney disease

## Abstract

Kidney disease is a significant health problem worldwide, affecting an estimated 10% of the global population. Kidney disease encompasses a diverse group of disorders that vary in their underlying pathophysiology, clinical presentation, and outcomes. These disorders include acute kidney injury (AKI), chronic kidney disease (CKD), glomerulonephritis, nephrotic syndrome, polycystic kidney disease, diabetic kidney disease, and many others. Despite their distinct etiologies, these disorders share a common feature of immune system dysregulation and metabolic disturbances. The immune system and metabolic pathways are intimately connected and interact to modulate the pathogenesis of kidney diseases. The dysregulation of immune responses in kidney diseases includes a complex interplay between various immune cell types, including resident and infiltrating immune cells, cytokines, chemokines, and complement factors. These immune factors can trigger and perpetuate kidney inflammation, causing renal tissue injury and progressive fibrosis. In addition, metabolic pathways play critical roles in the pathogenesis of kidney diseases, including glucose and lipid metabolism, oxidative stress, mitochondrial dysfunction, and altered nutrient sensing. Dysregulation of these metabolic pathways contributes to the progression of kidney disease by inducing renal tubular injury, apoptosis, and fibrosis. Recent studies have provided insights into the intricate interplay between immune and metabolic pathways in kidney diseases, revealing novel therapeutic targets for the prevention and treatment of kidney diseases. Potential therapeutic strategies include modulating immune responses through targeting key immune factors or inhibiting pro-inflammatory signaling pathways, improving mitochondrial function, and targeting nutrient-sensing pathways, such as mTOR, AMPK, and SIRT1. This review highlights the importance of the interplay between immune and metabolic pathways in kidney diseases and the potential therapeutic implications of targeting these pathways.

## 1. Introduction

Kidney disease is a significant health problem worldwide, affecting an estimated 10% of the global population [1]. The most common forms of kidney disease include chronic kidney disease (CKD) and acute kidney injury (AKI) [2]. However, kidney disease encompasses a diverse group of disorders that vary in their underlying pathophysiology, clinical presentation, and outcomes. These disorders include tubulointerstitial, glomerulonephritis, nephrotic syndrome, polycystic kidney disease, diabetic kidney disease, vascular disease, vasculitis, and congenital kidney disease, among others.

Despite significant advances in our understanding of the pathophysiology of kidney disease, current therapies remain limited and often ineffective [3,4]. Thus, the need for new therapeutic approaches to improve outcomes in patients with kidney disease is urgent. One promising area of research is the role of immunometabolism in the pathogenesis and progression of kidney disease [5,6].

Immunometabolism refers to the interplay between immune and metabolic pathways, which are tightly regulated in normal physiological conditions [7,8,9,10]. In pathological conditions, such as kidney disease, this delicate balance is disrupted, leading to immunometabolic dysregulation. Immunometabolic dysregulation involves various cell types, such as T cells, B cells, macrophages, and dendritic cells, as well as cytokines, chemokines, and metabolic processes, such as oxidative stress, mitochondrial dysfunction, and inflammation [8,11].

Recent studies have revealed a critical role for immunometabolic dysregulation in the pathogenesis of kidney disease [6,12,13]. Dysregulated immune responses and altered metabolic pathways interact in complex ways to contribute to the development and progression of kidney disease, regardless of the underlying etiology [8]. For instance, in CKD, chronic inflammation, oxidative stress, and altered lipid metabolism contribute to tubulointerstitial fibrosis and renal dysfunction [14,15,16,17,18]. Similarly, in diabetic kidney disease, hyperglycemia and dyslipidemia promote mitochondrial dysfunction and inflammation, leading to glomerular injury and renal fibrosis [19,20].

Given the profound impact of immunometabolic dysregulation on kidney disease outcomes, identifying new therapeutic targets to modulate these pathways is critical. In this review, we aim to provide a comprehensive overview of immunometabolic alterations in kidney disease, highlighting their clinical implications and potential therapeutic interventions. We discuss the most recent advancements in our understanding of the molecular mechanisms linking immunometabolism and kidney disease. Our review aims to provide insights into the critical role of immunometabolism in kidney disease, regardless of the underlying etiology, and its potential as a target for therapeutic intervention.

## 2. Immunometabolic Alterations in Kidney Disease

Immunometabolic alterations in kidney disease refer to the complex interplay between immune and metabolic pathways that are disrupted in pathological conditions [13,21]. These alterations involve various cell types, cytokines, chemokines, and metabolic processes, which together contribute to the pathogenesis and progression of kidney disease [8,22,23,24] (Figure 1).

T cells are an essential component of the adaptive immune response and play a crucial role in kidney disease [25,26]. In CKD, T-cell activation and infiltration contribute to chronic inflammation and renal fibrosis [27,28]. Activated T cells produce cytokines, such as IFN-γ and TNF-α, which promote inflammation and fibrosis in the kidney [29]. Additionally, T cells can directly induce tubular cell apoptosis and contribute to tubulointerstitial fibrosis [30,31]. In diabetic kidney disease, T cells also play a critical role in the pathogenesis of kidney disease [32,33]. T-cell infiltration in the glomerulus is associated with the development of albuminuria and renal fibrosis [16,34]. T cells in diabetic kidney disease also contribute to podocyte injury and the development of glomerular sclerosis [35,36].

B cells are another critical component of the adaptive immune response, and their role in kidney disease is becoming increasingly recognized [37]. In glomerulonephritis, autoantibodies produced by B cells play a significant role in the pathogenesis of the disease [38,39]. Autoantibodies can deposit in the glomerulus, leading to complement activation and subsequent inflammation and renal injury [40,41]. In diabetic kidney disease, B cells are also implicated in the development of the disease. B cells can produce pro-inflammatory cytokines and contribute to the infiltration of inflammatory cells in the kidney [42,43,44].

Macrophages are innate immune cells that play a critical role in the pathogenesis of human disease [45,46,47,48,49,50]. In CKD, macrophage infiltration in the kidney is associated with tubulointerstitial fibrosis and renal dysfunction [15,51,52]. Activated macrophages produce pro-inflammatory cytokines, such as TNF-α, IL-1β, and IL-6, which contribute to renal inflammation and fibrosis [53]. Macrophages can also promote renal fibrosis by producing TGF-β and promoting extracellular matrix deposition [54,55,56]. In diabetic kidney disease, macrophages contribute to the development of renal injury and fibrosis [57]. Macrophages are activated by advanced glycation end products (AGEs), leading to the production of pro-inflammatory cytokines and the promotion of renal fibrosis [57,58].

In addition to immune cell alterations, metabolic alterations also play a critical role in the pathogenesis of kidney disease. In CKD, oxidative stress and mitochondrial dysfunction are important metabolic alterations that contribute to renal injury and fibrosis [59,60]. Oxidative stress leads to the production of reactive oxygen species (ROS), which promote inflammation and fibrosis in the kidney [61]. Mitochondrial dysfunction can also lead to the production of ROS and promote renal fibrosis [62,63]. Additionally, altered lipid metabolism in CKD promotes tubulointerstitial fibrosis and renal dysfunction [64,65]. In diabetic kidney disease, hyperglycemia and dyslipidemia are the primary metabolic alterations that contribute to renal injury and fibrosis [65]. Hyperglycemia leads to the production of AGEs, which activate inflammatory cells and promote renal fibrosis [66,67]. Dyslipidemia leads to the accumulation of lipids in the kidney, promoting inflammation and fibrosis [68,69]. Furthermore, mitochondrial dysfunction in diabetic kidney disease contributes to the development of renal injury and fibrosis [70,71].

## 3. The Impact of Immunometabolic Dysregulation in Kidney Disease

### 3.1. Acute Kidney Injury (AKI)

Acute kidney injury (AKI) is a complex condition characterized by a rapid loss of renal function [72,73]. Immunometabolic dysregulation has been shown to play an important role in the pathogenesis of AKI [22,74]. This involves an imbalance between pro- and anti-inflammatory cytokines, leading to the activation of innate immune cells and subsequent tissue damage.

Several genes and pathways have been linked to immunometabolic dysregulation in AKI. One of the key pathways involved in the development of AKI is the hypoxia-inducible factor 1-alpha (HIF-1α) pathway [75,76]. Under hypoxic conditions, HIF-1α is stabilized and activates the transcription of genes involved in glycolysis, angiogenesis, and inflammation [77]. Studies have shown that HIF-1α plays a critical role in the development of AKI by promoting glycolysis in immune cells and contributing to the production of pro-inflammatory cytokines [78,79,80]. In addition, HIF-1α can also upregulate glucose transporter 1 (GLUT1), which facilitates glucose uptake in immune cells, and its upregulation has been linked to the development of AKI [80,81]. Moreover, recent studies have suggested that epigenetic modifications, such as DNA methylation and histone modifications, can contribute to the dysregulation of HIF-1α in AKI pathogenesis [82,83,84]. Another important gene involved in immunometabolic dysregulation in AKI is the gene encoding for inducible nitric oxide synthase (iNOS). iNOS is an enzyme that produces nitric oxide (NO), which is a potent regulator of immune cell function [85]. Dysregulation of iNOS has been implicated in the pathogenesis of AKI, with studies showing that iNOS-mediated NO production can contribute to tissue damage in the kidney [86,87,88].

In addition to HIF-1α and iNOS, toll-like receptors (TLRs) are involved in the recognition of pathogen-associated molecular patterns (PAMPs) and damage-associated molecular patterns (DAMPs), and their dysregulation has been linked to the development of AKI [89,90]. TLRs can activate nuclear factor kappa B (NF-κB), a transcription factor that regulates the expression of genes involved in inflammation and immune cell activation, and its dysregulation has been shown to contribute to the development of AKI [91,92]. The NLRP3 inflammasome, a multiprotein complex involved in the processing and secretion of pro-inflammatory cytokines, has also been implicated in the development of AKI. Studies have shown that NLRP3 inflammasome activation can contribute to the development of AKI by promoting the secretion of pro-inflammatory cytokines [93,94,95].

Furthermore, recent studies have shown that immunometabolic dysregulation in AKI also involves the dysregulation of lipid metabolism. For example, increased levels of free fatty acids (FFAs) can contribute to the development of AKI by activating inflammatory pathways in immune cells [96,97]. This process involves the activation of TLR4 and subsequent activation of NF-κB, resulting in the production of pro-inflammatory cytokines [98,99]. Moreover, dysregulation of the peroxisome proliferator-activated receptor gamma coactivator 1-alpha (PGC-1α), a transcriptional coactivator involved in the regulation of mitochondrial biogenesis and function [100], has been shown to contribute to the development of AKI by impairing mitochondrial function in immune cells [101,102,103]. Dysregulation of PGC-1α may also lead to the accumulation of ROS, which can cause oxidative stress and contribute to renal injury [104].

### 3.2. Chronic Kidney Disease (CKD)

Chronic kidney disease (CKD) is a progressive condition characterized by the gradual loss of kidney function over time. Dysregulation of immune cells and metabolism contribute to the accumulation of toxic metabolites, oxidative stress, and fibrosis, which are key contributors to the progression of CKD [105]. One of the key pathways involved in the development of CKD is dysregulated glucose metabolism in immune cells [106,107]. Studies have shown that this dysregulation can lead to the activation of pro-inflammatory pathways, oxidative stress, and endothelial dysfunction, all of which can contribute to the development of CKD [61]. GLUT1 and HIF-1α are two genes that have been implicated in the dysregulation of glucose metabolism in immune cells in the context of CKD [108,109]. Another important pathway involved in CKD is the activation of the NLRP3 inflammasome and subsequent cytokine production. Increased NLRP3 expression has been observed in patients with CKD, and inhibition of the NLRP3 inflammasome has been shown to ameliorate kidney damage in animal models of CKD [110]. Additionally, dysregulated lipid metabolism has been linked to the progression of CKD. Studies have shown that increased levels of FFAs can contribute to the development of CKD by activating inflammatory pathways and inducing oxidative stress [111,112]. In addition to the above-mentioned pathways, other genes involved in immune cell dysregulation in CKD include TLRs, NF-κB, and the renin–angiotensin–aldosterone system (RAAS). TLRs are involved in the recognition of PAMPs and DAMPs, and their dysregulation has been linked to the development of CKD [113,114,115]. NF-κB activation in CKD can be triggered by a variety of stimuli, including oxidative stress, hypoxia, and proinflammatory cytokines, such as TNF-α and IL-1β [116,117]. Furthermore, NF-κB activation is tightly linked to NLRP3 inflammasome activation in CKD. Activation of the NLRP3 inflammasome triggers the activation of NF-κB, which, in turn, leads to the production of more proinflammatory cytokines, creating a positive feedback loop that perpetuates the inflammatory response [118,119]. The RAAS is a hormone system that regulates blood pressure and fluid balance in the body, and its dysregulation has been linked to the development of CKD through its effects on renal hemodynamics and inflammation [120,121,122].

Furthermore, epigenetic modifications have been suggested to play a role in the dysregulation of genes involved in CKD pathogenesis [4]. For example, studies have shown that DNA methylation and histone modifications can contribute to the dysregulation of key genes involved in CKD, such as HIF-1α and NF-κB [123,124,125,126]. In conclusion, dysregulation of immune cells and metabolism can contribute to the pathogenesis and progression of CKD through various pathways and genes. Further research in this area may provide novel insights into the mechanisms underlying the development of CKD and help identify new therapeutic targets for the treatment of this condition.

#### 3.2.1. Lupus Nephritis

Lupus nephritis is a type of kidney inflammation that occurs as a result of systemic lupus erythematosus (SLE), an autoimmune disease [127]. Immunometabolic dysregulation is one of the key mechanisms underlying the pathogenesis of lupus nephritis [128]. Dysregulated metabolism in immune cells can contribute to the production of autoantibodies and the activation of inflammatory cells, leading to glomerular damage and renal dysfunction [129,130].

Several genes and pathways have been implicated in the dysregulated metabolism in immune cells in the context of lupus nephritis. One of the most studied pathways is the Warburg effect, which is characterized by the preferential use of glycolysis over oxidative phosphorylation in immune cells [131]. The upregulation of glycolysis is thought to be driven by various signaling pathways, including the phosphoinositide 3-kinase (PI3K)/Akt/mammalian target of rapamycin (mTOR) pathway, the HIF pathway, and the JAK/STAT pathway [132,133]. These pathways have been shown to contribute to the activation of immune cells and the production of autoantibodies in lupus nephritis [133,134,135,136].

The activation of the NLRP3 inflammasome is another key pathway involved in the pathogenesis of lupus nephritis, with the subsequent production of cytokines. The NLRP3 inflammasome contribute to tissue damage in lupus nephritis [137]. Studies have shown that the NLRP3 inflammasome is upregulated in lupus nephritis patients and that its inhibition can ameliorate kidney injury in animal models of lupus nephritis [137,138]. Moreover, dysregulated lipid metabolism has also been implicated in the pathogenesis of lupus nephritis. Studies have shown that increased levels of FFAs can contribute to the activation of immune cells and the production of autoantibodies in lupus nephritis [139,140]. The dysregulation of cholesterol metabolism has also been linked to the development of lupus nephritis. In addition to the above-mentioned pathways, other genes and pathways involved in the dysregulated metabolism in immune cells in lupus nephritis include TLRs, NF-κB, and the IFN pathway. TLRs are involved in the recognition of PAMPs and DAMPs, and their dysregulation has been linked to the activation of immune cells in lupus nephritis [141,142,143]. NF-κB is a transcription factor that regulates the expression of genes involved in inflammation and immune cell activation, and its dysregulation has been shown to contribute to the development of lupus nephritis [144,145]. The type I IFN pathway is another important pathway involved in the activation of immune cells in lupus nephritis, as the overexpression of type I IFN-inducible genes has been observed in lupus nephritis patients [146,147,148].

Furthermore, epigenetic modifications have also been suggested to play a role in the dysregulated metabolism in immune cells in lupus nephritis. For example, studies have shown that DNA methylation and histone modifications can contribute to the dysregulation of key genes involved in lupus nephritis, such as NF-κB [149]. In conclusion, immunometabolic dysregulation is a key mechanism underlying the pathogenesis of lupus nephritis.

#### 3.2.2. Diabetic Kidney Disease

Diabetic kidney disease is a common complication of diabetes mellitus and a leading cause of end-stage renal disease [150,151,152]. Dysregulated metabolism and inflammation are key factors in the pathogenesis of diabetic kidney disease. Impaired glucose metabolism leads to the accumulation of AGEs in the kidneys, which contribute to renal dysfunction and fibrosis [153]. GLUT1 and HIF-1α are two genes that have been implicated in the dysregulation of glucose metabolism in immune cells in the context of diabetic kidney disease [78,154].

In addition to dysregulated glucose metabolism, dysregulated lipid metabolism in immune cells has also been implicated in the pathogenesis of diabetic kidney disease. Studies have shown that increased levels of FFAs can contribute to the development of diabetic kidney disease by activating inflammatory pathways and inducing oxidative stress [155,156]. In particular, the peroxisome proliferator-activated receptor (PPAR) family of genes, which regulates lipid metabolism, has been shown to play a role in the pathogenesis of diabetic kidney disease [157,158]. The activation of the NLRP3 inflammasome and subsequent production of pro-inflammatory cytokines have been identified as critical drivers of diabetic kidney disease. The NLRP3 inflammasome is a multiprotein complex involved in the processing and secretion of pro-inflammatory cytokines, and its activation has been implicated in the development of diabetic kidney disease [159]. The inflammasome is activated by a variety of stimuli, including high glucose levels and the accumulation of AGEs [160]. The JAK/STAT signaling pathway is involved in many biological processes, including immune responses and inflammation, and has been implicated in the pathogenesis of diabetic kidney disease [161,162,163,164,165,166,167]. Studies have shown that the JAK/STAT pathway is activated in response to pro-inflammatory cytokines and growth factors, and its dysregulation can contribute to the progression of diabetic kidney disease [168]. The suppressor of cytokine signaling (SOCS) family of genes, which negatively regulates JAK/STAT signaling, has been shown to play a role in the development of diabetic kidney disease [169,170].

In conclusion, dysregulated metabolism and inflammation contribute to the development and progression of diabetic kidney disease through various pathways and genes, including dysregulated glucose and lipid metabolism, activation of the NLRP3 inflammasome, and dysregulated JAK/STAT signaling. Further research in this area may provide novel insights into the mechanisms underlying the development of diabetic kidney disease and help identify new therapeutic targets for the treatment of this condition.

#### 3.2.3. Polycystic Kidney Disease (PKD)

Immunometabolic dysfunction plays a critical role in the pathogenesis of PKD. Dysregulated metabolism in immune cells, such as the activation of the Warburg effect, has been implicated in the development and progression of PKD [171,172]. Additionally, studies have shown that immune cells in PKD exhibit increased mitochondrial stress and metabolic alterations, leading to impaired cellular energetics and increased oxidative stress [173].

One recent study has found that the inflammasome pathway, specifically the NLRP3 inflammasome, is activated in PKD, leading to the production of pro-inflammatory cytokines and subsequent cyst growth [174,175]. The activation of the NLRP3 inflammasome has been linked to the accumulation of damaged mitochondria and the release of mitochondrial DNA, which can trigger an inflammatory response in the kidney [23]. Another study has shown that PKD is associated with altered immune cell metabolism and an increased production of ROS. The authors suggest that this metabolic dysfunction may contribute to the activation of the NLRP3 inflammasome and the subsequent production of pro-inflammatory cytokines in PKD [174,176]. Furthermore, recent research has also linked PKD to dysregulated lipid metabolism in immune cells [61]. One study found that PKD is associated with altered lipid metabolism in T cells, leading to increased T-cell activation and subsequent inflammation in the kidney [177].

In summary, immunometabolic dysfunction, including dysregulated metabolism in immune cells, activation of the inflammasome pathway, altered mitochondrial function, and dysregulated lipid metabolism, contributes to the pathogenesis of PKD. These findings suggest that targeting immunometabolic pathways may provide a potential therapeutic strategy for PKD.

#### 3.2.4. Impact of Immunometabolic Dysregulation on Kidney Transplant Outcomes

Immunometabolic dysregulation has been increasingly recognized as an important contributor to kidney transplant outcomes. The immune response after kidney transplantation involves both the innate and adaptive immune systems, which interact with each other to establish a balance between tolerance and rejection [178,179]. Dysregulated metabolism and inflammation can disrupt this balance, leading to poor transplant outcomes, such as rejection, infection, and chronic allograft dysfunction [180,181].

One key pathway involved in immunometabolic dysregulation after kidney transplantation is the activation of the NLRP3 inflammasome. Studies have shown that activation of the NLRP3 inflammasome in both donor and recipient cells can contribute to the development of acute and chronic rejection [182]. Furthermore, activation of the NLRP3 inflammasome has also been implicated in the development of ischemia–reperfusion injury, a common complication during kidney transplantation [183,184,185]. Dysregulated metabolism in immune cells has also been implicated in poor kidney transplant outcomes. Specifically, the Warburg effect, a phenomenon where immune cells preferentially use glycolysis for energy production instead of oxidative phosphorylation, has been observed in both donor and recipient cells after kidney transplantation [186,187]. This metabolic switch has been associated with increased inflammation and oxidative stress, which can lead to allograft injury and rejection [188,189]. Finally, dysregulation of lipid metabolism in immune cells has also been implicated in poor kidney transplant outcomes [190]. Studies have shown that high levels of triglycerides and low levels of high-density lipoprotein (HDL) cholesterol are associated with an increased risk of acute rejection and chronic allograft dysfunction [191,192]. Dysregulated lipid metabolism in immune cells can also lead to the production of pro-inflammatory cytokines and the activation of the NLRP3 inflammasome [193,194].

In conclusion, immunometabolic dysregulation plays a critical role in kidney transplant outcomes. Dysregulated metabolism and inflammation can disrupt the delicate balance between tolerance and rejection, leading to poor transplant outcomes, such as rejection, infection, and chronic allograft dysfunction. Understanding the mechanisms underlying immunometabolic dysregulation in kidney transplantation may lead to the development of novel therapeutic strategies to improve transplant outcomes.

## 4. Potential Therapeutic Interventions Targeting Immunometabolism in Kidney Disease

Immunometabolic dysregulation is a promising target for the development of novel therapeutic interventions for kidney disease. Several current and emerging therapies targeting immunometabolism have shown promising results in preclinical and clinical studies.

One potential therapeutic intervention is targeting the NLRP3 inflammasome, a key component of the innate immune system that plays a role in the activation of pro-inflammatory cytokines. The NLRP3 inflammasome, a multimeric protein complex, acts as a key regulator of innate immunity and exhibits predominant expression within diverse renal cell populations, encompassing tubular epithelial cells, glomerular cells, and infiltrating immune cells within the kidney [195,196]. The NLRP3 inflammasome can be activated in response to different signals, such as PAMPs, DAMPs, and oxidized mitochondrial DNA fragments. Once activated, the NLRP3 inflammasome triggers the production and release of pro-inflammatory cytokines, particularly IL-1β and IL-18, leading to an amplified inflammatory response within the renal microenvironment [197,198]. The significance of NLRP3 inflammasome activation in renal diseases lies in its contribution to the pathogenesis and progression of various renal conditions [93,199]. Persistent or dysregulated activation of the NLRP3 inflammasome has been implicated in the development of glomerulonephritis, diabetic nephropathy, tubulointerstitial nephritis, and other inflammatory renal disorders. The released pro-inflammatory cytokines, IL-1β and IL-18, promote immune cell recruitment, exacerbate tissue damage, and stimulate fibrotic responses in the kidney [200,201,202]. Moreover, the NLRP3 inflammasome can modulate the activation and function of other inflammatory signaling pathways, such as NF-κB and mitogen-activated protein kinases (MAPKs), amplifying the inflammatory cascade in renal diseases [203]. Furthermore, the influence of the NLRP3 inflammasome extends beyond inflammation, as it has been implicated in regulating renal cell death pathways. Activation of the NLRP3 inflammasome can lead to pyroptosis, a highly inflammatory form of cell death characterized by releasing pro-inflammatory cytokines and forming membrane pores [204]. Pyroptosis of renal cells can exacerbate tissue injury and contribute to the loss of renal function [205]. Additionally, the NLRP3 inflammasome has been associated with the activation of other cell death mechanisms, including apoptosis and necroptosis, further highlighting its involvement in renal disease pathogenesis [93]. Inhibitors of the NLRP3 inflammasome, such as MCC950 and CY-09, have been shown to ameliorate renal injury and improve kidney function in various animal models of kidney disease [206,207,208]. However, the clinical efficacy of these inhibitors remains to be tested in human trials. Another potential therapy is the modulation of the Warburg effect, a metabolic alteration characterized by enhanced glycolysis and reduced oxidative phosphorylation. Targeting the Warburg effect in immune cells has shown potential in the treatment of kidney disease. For instance, the use of the glycolysis inhibitor 2-deoxyglucose (2-DG) has been shown to reduce renal injury and inflammation in animal models of kidney disease [171,209,210]. Additionally, several other inhibitors of glycolysis, such as dichloroacetate (DCA) and lonidamine, are currently under investigation as potential therapies for kidney disease [211,212].

In addition to targeting specific pathways, several emerging therapies aim to modulate the overall metabolic state of immune cells in kidney disease. One example is the use of metformin, a widely used drug for the treatment of diabetes, which has been shown to have immunomodulatory effects [213]. Preclinical studies have demonstrated the potential of metformin in reducing renal injury and inflammation in models of kidney disease. Similarly, the use of rapamycin, an inhibitor of the mammalian target of rapamycin (mTOR), has been shown to have immunosuppressive and renoprotective effects in various animal models of kidney disease [214]. Cyclosporine A is an immunosuppressant commonly used in renal transplant patients to prevent rejection by inhibiting immune system activity and reducing inflammatory responses and immune-mediated kidney damage. While its primary focus is on the immune system, Cyclosporine A may also have some impact on metabolic processes [215]. Glucocorticoids, such as prednisolone, possess anti-inflammatory properties and are frequently prescribed for various kidney diseases, mitigating inflammation and immune-mediated injury. They can also affect metabolism by influencing glucose metabolism and lipid metabolism [216]. Angiotensin-converting enzyme inhibitors (ACEIs) and angiotensin receptor blockers (ARBs) are widely employed in managing hypertension and kidney disease. In addition to their blood-pressure-lowering effects, these medications can have an impact on metabolic processes, including the regulation of blood glucose levels and lipid metabolism [217]. It is important to consult healthcare professionals for personalized treatment decisions, taking into account the specific condition and needs of each patient.

While these immunometabolic therapies hold promise, there are also potential limitations and concerns to consider. For instance, the modulation of immune cell metabolism may have unintended consequences on other metabolic pathways and cellular functions. Additionally, the long-term safety and efficacy of these therapies in humans remain to be established. In conclusion, targeting immunometabolism represents a promising approach for the development of novel therapies for kidney disease. While several therapies have shown promise in preclinical and clinical studies, further research is needed to fully establish their safety and efficacy in humans.

## 5. Future Directions for Research in Immunometabolism and Kidney Disease

Despite significant progress in understanding the role of immunometabolism in kidney disease, there are still many gaps in our knowledge. Here, we outline some areas of needed research to better understand the complex interactions between immunometabolism and kidney disease.

1. Elucidating the mechanisms of immunometabolic dysregulation in kidney disease: While the role of immunometabolism in kidney disease is becoming increasingly clear, the specific molecular and cellular mechanisms underlying this dysregulation are still not fully understood. Future research should focus on elucidating these mechanisms to better understand how immunometabolic dysregulation contributes to kidney disease pathogenesis.

2. Identifying novel immunometabolic targets for therapeutic interventions: While current and emerging immunometabolic therapies for kidney disease show promise, there is a need for the identification of additional immunometabolic targets for therapeutic interventions. Innovative approaches and technologies, such as multi-omics and single-cell analysis, may help identify new targets and pathways involved in immunometabolic dysregulation.

3. Personalizing immunometabolic therapies for kidney disease: The heterogeneity of kidney disease suggests that personalized therapeutic approaches may be necessary. Future research should aim to identify specific patient subgroups that may benefit from certain immunometabolic therapies, as well as develop biomarkers to predict treatment response.

## 6. Conclusions

This review highlights the significant role of immunometabolic dysregulation in kidney disease. The interplay between immune and metabolic pathways affects the development and progression of various kidney diseases, including AKI, CKD, lupus nephritis, diabetic kidney disease, PKD, and kidney transplant outcomes. Potential therapeutic interventions targeting immunometabolism show promise, but further research is needed. Understanding the interactions between immune and metabolic processes is crucial for future advancements in treating kidney disease.

## Figures and Tables

**Figure 1 cells-12-01584-f001:**
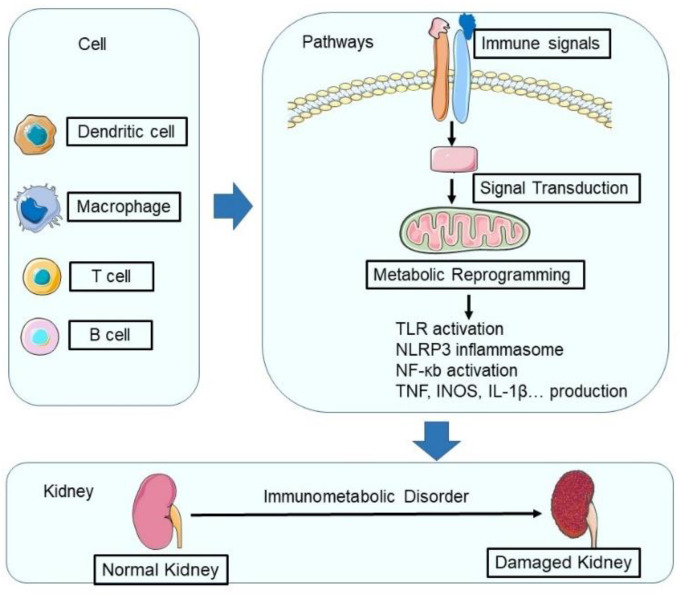
During renal injury, the metabolic programming of immune cells undergoes significant changes. In a healthy kidney, macrophages use α-ketoglutarate derived from glutamine to maintain their phenotypes, while both resident macrophages and T lymphocytes rely on oxidative phosphorylation (OXPHOS). However, during renal injury, hypoxia-inducible factor-1α (HIF-1α)-mediated metabolic reprogramming occurs, leading to increased glycolysis and altered amino acid metabolism in immune cells. In addition, the activation of innate pattern recognition receptors, such as Toll-like receptors (TLRs), NOD-like receptors (NLRs), and inflammasomes, triggers intracellular pathways that converge on nuclear factor κB (NF-κB), resulting in the production of pro-inflammatory cytokines (such as tumor necrosis factor (TNF) and interleukin-1β (IL-1β)) and chemokines. This complex network of metabolic and inflammatory responses ultimately contributes to the progression of renal injury and disease. Elements of some figures were produced using Servier Medical Art, https://smart.servier.com.

## Data Availability

No new data were created or analyzed in this study. Data sharing is not applicable to this article.

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
