# Peer review of "The Interplay between Immune and Metabolic Pathways in Kidney Disease"

_cells, 2023, doi:10.3390/cells12121584_

Round 1
Reviewer 1 Report
This manuscript is a review article in which the authors describe some metabolic and immune pathways in kidney disease. At the end of the article, some potential therapeutic options for targeting impaired metabolic pathways are discussed.
This is a good review paper that will be of interest to readers in the field of nephrology.
I have the following comments and questions for the authors:
1. In the first paragraph, glomerulonephritis, polycystic kidney disease, nephrotic syndrome, and diabetic kidney disease are mentioned among the diseases. As we know, there are many other diseases in nephrology, such as tubulointerstitial, vascular disease, vasculitis, congenital kidney disease, and many others. Please explain.
2. In section 3, the authors analysed imunometabolic dysregulation only for acute kidney injury, chronic kidney disease, lupus nephritis, diabetic nephropathy (the term diabetic kidney disease is more appropriate), polycystic kidney disease, and renal transplantation. Similar to question 1, I would like to ask the authors to explain what criteria were used to select exactly these diseases.
3. Figure 1 is not mentioned in the text and it is not known with which part of the text in the manuscript it is associated.
4. Activation of the NLRP3 inflammasome is involved in the inflammatory response in almost all renal diseases described in this review article. It would be appropriate to write a separate paragraph on the precise role, significance, and influence of this multimeric protein complex.
Reviewer 2 Report
Kidney disease is a signiffcant health problem worldwide, affecting an estimated 10% of the global population. Kidney disease encompasses a diverse group of disorders that vary in their underlying pathophysiology, clinical presentation, and outcomes. Despite their distinct etiologies, these disorders share a common feature of immune system dysregulation and metabolic disturbances. Since the immune system and metabolic pathways are intimately connected and interact to modulate the pathogenesis of kidney diseases, in this article, Qu et al., reviewed that the intricate interplay between immune and metabolic pathways in kidney diseases. The article was well written and developed based on a large of number of published articles. However, it seems that most of articles cited are old. The authors need to cite more articles published in the past five years. In addition, the authors should describe more about medications or inhibitors that can interferes both immune system and metabolic pathways in addition to metformin.
well written
